# Constrained Proximal Policy Optimization

## Abstract

The problem of constrained reinforcement learning (CRL) holds significant importance as it provides a framework for addressing critical safety satisfaction concerns in the field of reinforcement learning (RL). However, with the introduction of constraint satisfaction, the current CRL methods necessitate the utilization of second-order optimization or primal-dual frameworks with additional Lagrangian multipliers, resulting in increased complexity and inefficiency during implementation. To address these issues, we propose a novel first-order feasible method named Constrained Proximal Policy Optimization (CPPO). By treating the CRL problem as a probabilistic inference problem, our approach integrates the Expectation-Maximization framework to solve it through two steps: 1) calculating the optimal policy distribution within the feasible region (E-step), and 2) conducting a first-order update to adjust the current policy towards the optimal policy obtained in the E-step (M-step). We establish the relationship between the probability ratios and KL divergence to convert the E-step into a convex optimization problem. Furthermore, we develop an iterative heuristic algorithm from a geometric perspective to solve this problem. Additionally, we introduce a conservative update mechanism to overcome the constraint violation issue that occurs in the existing feasible region method. Empirical evaluations conducted in complex and uncertain environments validate the effectiveness of our proposed method, as it performs at least as well as other baselines.

## 1 Introduction

In recent years, reinforcement learning (RL) has achieved huge success in various aspects (Le et al., 2022; Li et al., 2022; Silver et al., 2018), especially in the field of games. However, due to the increased safety requirements in practice, researchers are starting to consider the constraint satisfaction in RL. Compared with unconstrained RL, constrained RL (CRL) incorporates certain constraints during the process of maximizing cumulated rewards, which provides a framework to model several important topics in RL, such as safe RL (Paternain et al., 2022), highlighting the importance of this problem in industrial applications.

The current methods for solving the CRL problem can be mainly classified into two categories: primal-dual method (Paternain et al., 2022; Stooke et al., 2020; Zhang et al., 2020; Altman, 1999) and feasible region method (Achiam et al., 2017; Yang et al., 2020). The primal-dual method introduces the Lagrangian multiplier to convert the constrained optimization problem into an unconstrained dual problem by penalizing the infeasible behaviours, promising the CRL problem to be resolved in a first-order manner. Despite the primal-dual framework providing a way to solve CRL in first-order manner, the update of the dual variable, i.e., the Lagrangian multiplier, tends to be slow and unstable, affecting the overall convergent speed of the algorithms. In contrast, the feasible region method provides a faster learning method by introducing the concept of the feasible region into the trust region method. With either searching in the feasible region (Achiam et al., 2017) or projecting into

the feasible region (Yang et al., 2020), the feasible region method can guarantee the generated policies stay in the feasible region. However, the introduction of the feasible region in the proposed method relies on computationally expensive second-order optimization using the inverse Fisher information matrix. This approach can lead to inaccurate estimations of the feasible region and potential constraint violations, as reported in previous studies (Ray et al., 2019).

To address the existing issues mentioned above, this paper proposed the Constrained Proximal Policy Optimization (CPPO) algorithm to solve the CRL problem in a first-order, easy-to-implement way. CPPO employs a two-step Expectation-Maximization approach to solve the problem by firstly calculating the optimal policy (E-step) and then conducting a first-order update to reduce the distance between the current policy and the optimal policy (M-step), eliminating the usage of the Lagrangian multiplier and the second-order optimization. The main contributions of this work are summarized as follows:

- To our best knowledge, the proposed method is the first **first-order feasible region method** without using dual variables or second-order optimization, which significantly reduces the difficulties in tuning hyperparameters and the computing complexity.

- An **Expectation-Maximization** (EM) framework based on **advantage value** and **probability ratio** is proposed for solving the CRL problem efficiently. By converting the CRL problem into a probabilistic inference problem, the CRL problem can solved in first order manner without dual variables.

- To solve the convex optimization problem in E-step, we established the relationship between the probability ratios and KL divergence, and developed an **iterative heuristic algorithm** from a geometric perspective.

- A **recovery update** is developed when the current policy encounters constraint violation. Inspired by Bang-bang control, this update strategy can improve the performance of constraint satisfaction and reduce the switch frequency between normal update and recovery update.

- The proposed method is evaluated in several benchmark environments. The results manifest its comparable performance over other baselines in complex environments.

This paper is organized as follows. Section 2 introduces the concept of constrained markov decision process and present an overview of related works in the field. The Expectation-Maximization framework and the technical details about the proposed constrained proximal policy optimization method are proposed in Section 3. Section 4 verifies the effectiveness of the proposed method through several testing scenarios and an ablation study is conducted to show the effectiveness of the proposed recovery update. Section 5 states the limitations and the boarder impact of the proposed method. Finally, a conclusion is drawn in Section 6.

## 2 Preliminary and Related Work

### 2.1 Constrained Markov Decision Process

Constrained Markov Decision Process(CMDP) is a mathematical framework for modelling decision-making problems subjected to a set of cost constraints. A CMDP can be defined by a tuple $(\mathcal{S}, \mathcal{A}, P, r, \gamma, \mu, C)$, where $\mathcal{S}$ is the state space, $\mathcal{A}$ is the action space, $P : \mathcal{S} \times \mathcal{A} \times \mathcal{S} \to (0, 1)$ is the transition kernel, $r : \mathcal{S} \times \mathcal{A} \to \mathbb{R}$ is the reward function, $\gamma \to (0, 1)$ is the discount factor, $\mu : \mathcal{S} \to (0, 1)$ is the initial state distribution, and $C := \{c_i \in C \mid c_i : \mathcal{S} \times \mathcal{A} \to \mathbb{R}, i = 1, 2, \ldots, m\}$ is the set of $m$ cost functions. For simplicity, we only consider a CRL problem with one constraint in the following paper and use $c$ to represent the cost function. Note that, although we restrict our discussion to the case with only one constraint, the method proposed in this paper can be naturally extended to the multiple constraint case. However, the result may not as elegant as the one constraint case. Compared with the common Markov Decision Process(MDP), CMDP introduces a constraint on the cumulated cost to restrict the agent's policies. Considering a policy $\pi(s \mid a) : \mathcal{S} \times \mathcal{A} \to (0, 1)$, the goal of MDP is to find the $\pi$ that maximizes the expected discounted returns $J_r(\pi) = \mathbb{E}_\tau \left[ \sum_{t=0}^{\infty} \gamma^t r(s_t) \right]$, where $\tau$ is the trajectories generated based on $\pi$. Based on these settings, CMDP applied a threshold $d$ on the expected discounted cost returns $J_c(\pi) = \mathbb{E}_\tau \left[ \sum_{t=0}^{\infty} \gamma^t c(s_t) \right]$. Thus, the CMDP problem can be formed as finding policy $\pi^*$ that $\pi^* = \text{argmax}_\pi J_r(\pi)$ s.t. $J_c(\pi^*) \leq d$. The advantage function $A$ and the cost advantage function $A_c$ is defined as $A(s_t, a_t) = Q(s_t, a_t) - V(s_t)$

and $A_c(s_t, a_t) = Q_c(s_t, a_t) - V_c(s_t)$ where $Q(s_t, a_t) = \mathbb{E}_\tau \left[ \sum_{t=0}^\infty \gamma^t r \mid s_0 = s_t, a_0 = a_t \right]$ and $V(s_t) = \mathbb{E}_\tau \left[ \sum_{t=0}^\infty \gamma^t r \mid s_0 = s_t \right]$ are the corresponding Q-value and V-value for reward function, and $Q_c(s_t, a_t) = \mathbb{E}_\tau \left[ \sum_{t=0}^\infty \gamma^t c \mid s_0 = s_t, a_0 = a_t \right]$ and $V_c(s_t) = \mathbb{E}_\tau \left[ \sum_{t=0}^\infty \gamma^t c \mid s_0 = s_t \right]$ are the corresponding Q-value and V-value for cost function. Note that both $A$ and $A_c$ in the batch are centered to moves theirs mean to 0, respectively.

## 2.2 Related Work

### 2.2.1 Proximal Policy Optimization (PPO)

Proximal policy optimization (PPO) (Schulman et al., 2017) is a renowned on-policy RL algorithm for its stable performance and easy implementation. Based on the first-order optimization methodology, PPO addresses the challenge of the unconstrained RL problem through the surrogate objective function that proposed in Trust Region Policy Optimization (TRPO) (Schulman et al., 2015a). With the clipping and early stop trick, PPO can keep the new policy to stay within the trust region. Thanks to its stability and superior performance, the PPO algorithm has been employed in various subfields of RL like multi-agent RL (Yu et al., 2021), Meta-RL (Yu et al., 2020). However, due to the extra constraint requirements, the direct application of PPO in CRL problems is not feasible. The extra constraint requirements cause PPO not only restricted by the trust region but also the constraint feasible region, which significantly increases the challenge in conducting first-order optimization. Despite the difficulties in the direct application of PPO in CRL, researchers are still searching for a PPO-like method to solve CRL problems with stable and superior performance.

### 2.2.2 Constrained Reinforcement Learning

The current methods for solving the CRL problem can be mainly divided into two categories: primal-dual method (Paternain et al., 2022; Stooke et al., 2020; Zhang et al., 2020) and feasible region method (Achiam et al., 2017; Yang et al., 2020). The primal-dual method converts the original problem into a convex dual problem by introducing the Lagrangian multiplier. By updating the policy parameters and Lagrangian multiplier iteratively, the policies obtained by the primal-dual method will gradually converge towards a feasible solution. However, the usage of the Lagrange multiplier introduces extra hyperparameters into the algorithm and slows down the convergence speed of the algorithm due to the characteristic of the integral controller. Stooke et al. (2020) tries to solve this issue by introducing PID control into the update of the Lagrangian multiplier, but this modification will introduce more hyperparameters and cause the algorithm to be complex. Different from the primal-dual method, the feasible region method estimates the feasible region within the trust region using linear approximation and subsequently determines the new policy based on the estimated feasible region. A representative method is constrained policy optimization (CPO). By converting the CRL to a quadratically constrained linear program, CPO (Achiam et al., 2017) can solve the problem efficiently. However, the uncertainties inside the environment may cause an inaccurate cost assessment, which will affect the estimation of the feasible region and cause the learned policy to fail to meet the constraint requirements, as shown in Ray et al. (2019). Another issue of CPO is that it uses the Fisher information matrix to estimate the KL divergence in quadratic approximation, which is complex in computing and inflexible in network structure.

To address the second-order issue in CRL, several researchers (Zhang et al., 2020; Liu et al., 2022) proposed the EM-based algorithm in a first-order manner. FOCOPS (Zhang et al., 2020) obtain the optimal policy from advantage value, akin to the maximum entropy RL, and perform a first-order update to reduce the KL divergence between the current policy and the optimal policy. Despite its significant improvement in performance compared to CPO, FOCOPS still necessitates the use of a primal-dual method to attain a feasible optimal policy, which introduces a lot of hyperparameters for tuning, resulting in a more complex tuning process. CVPO (Liu et al., 2022) extends the maximum a posteriori policy optimization (MPO) (Abdolmaleki et al., 2018) method to the CRL problem, allowing for the efficient calculation of the optimal policy from Q value in an off-policy manner. However, this algorithm still requires the primal-dual framework in optimal policy calculation and necessitates additional samplings during the training, increasing the complexity of implementation. Thus, the development of a simple-to-implement, first-order algorithm with superior performance, remains a foremost goal for researchers in the CRL subfield.

# 3 Constrained Proximal Policy Optimization (CPPO)

As mentioned in Section 2, existing CRL methods often require second-order optimization for feasible region estimation or the use of dual variables for cost satisfaction. These approaches can be computationally expensive or result in slow convergence. To address these challenges, we proposed a two-step approach in an EM fashion named Constrained Proximal Policy Optimization (CPPO), the details will be shown in this section.

## 3.1 Modelling CRL as Inference

Instead of directly pursuing an optimal policy to maximize rewards, our approach involves conceptualizing the problem of Constrained Reinforcement Learning (CRL) as a probabilistic inference problem. This is achieved by assessing the reward performance and constraint satisfaction of state-action pairs and subsequently increasing the likelihood of those pairs that demonstrate superior reward performance while adhering to the constraint requirement. Suppose the event of state-action pairs under policy $\pi_\theta$ can maximize reward is represented by optimality variable $O$, we assume the likelihood of state-action pairs being optimal is proportional to the exponential of its advantage value: $p(O = 1|(s,a)) \propto \exp(A(s,a)/\alpha)$ where $\alpha$ is a temperature parameter. Denote $q(a \mid s)$ is the feasible posterior distribution estimated from the sampled trajectories under current policy $\pi$, $p_\pi(a \mid s)$ is the probability distribution under policy $\pi$, and $\theta$ is the policy parameters. We can have following evidence lower bound(ELBO) $\mathcal{J}(q, \theta)$ using surrogate function(see Appendix B for detailed proof)

$$\log p_{\pi_\theta}(O = 1) \geq \mathbb{E}_{s\sim d^\pi, a\sim\pi} \left[ \frac{q(a|s)}{p_\pi(a|s)} A(s,a) \right] - \alpha D_{\text{KL}}(q \parallel \pi_\theta) + \log p(\theta) = \mathcal{J}(q, \theta), \quad (1)$$

where $d^\pi$ is the state distribution under current policy $\pi$, $p(\theta)$ is a prior distribution of policy parameters. Considering $q(a \mid s)$ is a feasible policy distribution, we also have following constraint (Achiam et al., 2017)

$$J_c(\pi) + \frac{1}{1 - \gamma} \mathbb{E}_{s\sim d^\pi, a\sim\pi} \left[ \frac{q(a|s)}{p_\pi(a|s)} A_c(s,a) \right] \leq d, \quad (2)$$

where $d$ is the cost constraint. By performing iterative optimization of the feasible posterior distribution $q$ (E-step) and the policy parameter $\theta$ (M-step), the lower bound $\mathcal{J}(q, \theta)$ can be increased, resulting in an enhancement in the likelihood of state-action pairs that have the potential to maximize rewards.

## 3.2 E-Step

### 3.2.1 Surrogate Constrained Policy Optimization

As mentioned in the previous section, we will firstly optimize the feasible posterior distribution $q$ to maximize ELBO in E-step. The feasible posterior distribution $q$ plays a crucial role in determining the upper bound of the ELBO since the KL divergence is non-negative. Consequently, $q$ needs to be theoretically optimal to maximize the ELBO. By converting the soft KL constraint in Equation (1) into a hard constraint and combining the cost constraint in Equation (2),the optimization problem of $q$ can be expressed as follows:

$$\begin{aligned} \underset{q}{\text{maximize}} \quad & \mathbb{E}_{s\sim d^\pi, a\sim\pi} \left[ \frac{q(a|s)}{p_\pi(a|s)} A(s,a) \right] \\ \text{s.t.} \quad & J_c(\pi) + \frac{1}{1 - \gamma} \mathbb{E}_{s\sim d^\pi, a\sim\pi} \left[ \frac{q(a|s)}{p_\pi(a|s)} A_c(s,a) \right] \leq d, \ D_{\text{KL}}(q \parallel \pi) \leq \delta, \end{aligned} \quad (3)$$

where $\delta$ is the reverse KL divergence constraint that determine the trust region. During the E-step, it is important to note that the optimization is independent of $\theta$, meaning that the policy $\pi_\theta$ remains fixed to the current sampled policy $\pi$. Even we know the closed-form expression of $p_{\pi_\theta}$, it is impractical to solve the closed-form expression of $q$ from Equation (3), as we still needs the closed-form expression of $d^\pi$ for calculating. Therefore, we we opt to represent the solution of $q$ in a non-parametric manner by calculating the probability ratio $v = \frac{q(a|s)}{p_\pi(a|s)}$ for the sampled state-action pairs, allowing us to

avoid explicitly parameterizing $q$ and instead leverage the probability ratio to guide the optimization process. After relaxing the reverse KL divergence constraint with the estimated reverse KL divergence calculated through importance sampling, we can obtain

$$\underset{v}{\text{maximize}} \quad \mathbb{E}_{s \sim d^{\pi}, a \sim \pi} \left[ v A(s, a) \right]$$
$$\text{s.t.} \quad \mathbb{E}_{s \sim d^{\pi}, a \sim \pi} \left[ v A_c(s, a) \right] \leq d' \tag{4}$$
$$\underset{\substack{s \sim d^{\pi} \\ a \sim \pi}}{\mathbb{E}} \left[ v \log v \right] \leq \delta.$$

where $d'$ the scaled cost margin $d' = (1 - \gamma)(d - J_c(\pi))$. Although Equation (4) is convex optimization problem that can be directly solved through existing convex optimization algorithm, the existence of non-polynomial KL constraint tends to cause the optimization to be computationally expensive. To overcome this issue, the following proposition is proposed to relax Equation (4) into an linear optimization problem with quadratic constraint.

**Proposition 3.1.** *Denote $v$ as the probability ratios $\frac{q(a|s)}{p_{\pi}(a|s)}$ calculated from sampled trajectories. If there are a sufficient number of sampled $v$, we have $\mathbb{E}[v] = 1$ and $\mathbb{E}\left[v \log v\right] \leq Var(v - 1)$.*

With Proposition 3.1, the relationship between reverse KL divergence and $l^2$-norm of vector $v - 1$ is constructed. Also, consider that the expectation of $v$ equals 1, the optimization variable can be changed from $v$ to $v - 1$. Let $\overline{v}$ denote the vector consists of $v - 1$ and replace the reverse KL divergence constraint with the $l^2$-norm constraint, Equation (4) can be rewritten in the form of vector multiplication

$$\underset{\overline{\mathbf{v}}}{\text{maximize}} \quad \overline{\mathbf{v}} \cdot \mathbf{A}$$
$$\text{s.t.} \quad \overline{\mathbf{v}} \cdot \mathbf{A}_c \leq N d', \ \|\overline{\mathbf{v}}\|_2 \leq 2N\delta' \tag{5}$$
$$\mathbb{E}(\overline{\mathbf{v}}) = 0, \ \overline{\mathbf{v}} > -1 \text{ element-wise,}$$

where $\mathbf{A}$ and $\mathbf{A}_c$ are the advantage value vectors for reward and cost (for all sampled state-action pairs in one rollout) respectively, $N$ is the number of state-action pair samples, $\delta'$ is $l^2$-norm constraint, and the element-wise lower bound of $\overline{\mathbf{v}}$ is $-1$, as $v > 0$. Thus, the optimal feasible posterior distribution $q$ expressed through $\overline{\mathbf{v}}$ can be obtained by solving the aforementioned optimization problem.

*Remark* 3.2. By replacing the non-polynomial KL constraint with an $l^2$-norm constraint, the original optimization problem in Equation (4) can be reformulated as a geometric problem. This reformulation enables the use of the proposed heuristic method to efficiently solve the problem **without the need for dual variables**.

*Remark* 3.3. Our proposed method builds upon the idea presented in CVPO (Liu et al., 2022) of treating the CRL problem as a probabilistic inference problem. However, our approach improves upon their idea in two significant ways. Firstly, the probabilistic inference problem in our method is constructed based on **advantage value**, which is more effective in reducing the bias in estimating the cost return, compared to the Q-value used in CVPO. Secondly, while CVPO tries to directly calculate the value of $q(a|s)$, our method employs the **probability ratio** $v$ to represent $q$. By replacing $q(a|s)$ with $v$, our method only needs to find a vector of $v$ whose elements are positive and $\mathbb{E}[v] = 1$, thereby negating the need to sample multiple actions in one state to calculate the extra normalizer that ensures $q$ is a valid distribution. This results in a significant reduction in computational complexity.

### 3.2.2 Recovery update

Although the optimal solution $q$ in Section 3.2.1 is applicable when the current policy is out of the feasible region, the inconsistent between optimal $q$ and $\pi_{\theta}$ and the inaccurate cost evaluations tends to result in the generation of infeasible policies, as demonstrated in Ray et al. (2019) where CPO fail to satisfy constraint. To overcome this issue, a recovery update strategy is proposed for pushing the agent back to the feasible region. This strategy aims to minimize costs while preserving or minimizing any reduction in overall reward return. In the event that it is not possible to recover from the infeasible region without compromising the reward return, the strategy aims to identify an optimal policy within the feasible region that minimizes the adverse impact on the reward return. The optimization problem in recovery update can be expressed as

$$\text{if } \overline{\mathbf{v}} \cdot \mathbf{A} \geq 0 \text{ not exists when } \overline{\mathbf{v}} \cdot \mathbf{A}_c \leq N d': \quad \underset{\overline{\mathbf{v}}}{\text{maximize}} \ \overline{\mathbf{v}} \cdot \mathbf{A}$$
$$\text{else:} \quad \underset{\overline{\mathbf{v}}}{\text{minimize}} \ \overline{\mathbf{v}} \cdot \mathbf{A}_c \tag{6}$$
$$\text{s.t.} \quad \|\overline{\mathbf{v}}\|_2 \leq 2N\delta', \ \mathbb{E}(\overline{\mathbf{v}}) = 0, \ \overline{\mathbf{v}} > -1 \text{ element-wise.}$$

Figure 1 illustrates the recovery update strategy from the perspective of geometry. The blue, red, and yellow arrows represent the direction of minimizing cost, maximizing reward and the recovery update, respectively. The reward preservation region is defined by the zero reward boundary, which is depicted as the dashed line perpendicular to the red arrow. As a result, the semi-circle encompassing the red arrow indicates a positive increment in reward. Case 1 and Case 3 illustrate the case when the reward preservation region has an intersection with the feasible region. In these cases, we choose the direction of minimizing cost within the reward preservation region, e.g., the recovery update direction is coincident with the dashed line in Case 1, and the recovery update direction is coincident with the blue arrow in Case 3. Case 2 shows the case when there is no intersection between the reward preservation region and the feasible region. In this case, the direction with the least damage to reward is chosen. If we use an angle $\alpha$ to represent the direction of update, then we can have $\alpha = \mathrm{Clip}(\alpha, \max(\theta_f, \theta_A + \pi/2), \pi)$, where $\theta_A$ represents the direction of $\mathbf{A}$, $\theta_f$ is the minimum angle that can point toward the feasible region.

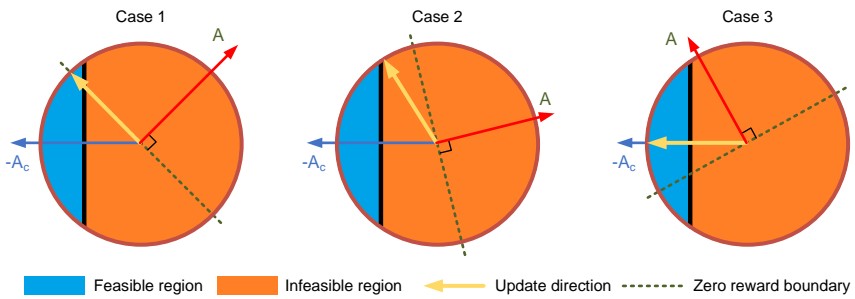

Figure 1: The illustration of recovery update.

To further improve the constraint satisfaction performance, a switching mechanism inspired by bang-bang control (Lasalle, 1960) is introduced. As shown in Figure 2, the agent will initially conduct normal update in Section 3.2.1; when the agent violates the cost constraint, it will switch to recovery update to reduce the cost until the cost is lower than the lower switch cost. By incorporating this switching mechanism, a margin is created between the lower switch cost and the cost constraint. This margin allows for a period of normal updates before the recovery update strategy is invoked. As a result, this mechanism prevents frequent switching between the two strategies, leading to improved performance in both reward collection and cost satisfaction. This switching mechanism effectively balances the exploration of reward-maximizing actions with the need to maintain constraint satisfaction.

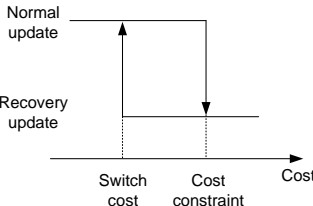

Figure 2: The switch mechanism inspired by bang-bang control. Once the current policy violates the cost constraint, the agent will switch to recovery update until it reaches the switch cost.

### 3.3 Heuristic algorithm from geometric interpretation

Section 3.2 and Section 3.4 provide a framework for solving CRL problem in theory. However, solving Equation (5) and Equation (6) in Section 3.2 is a tricky task in practice. To reduce the computation complexity, an iterative heuristic algorithm is proposed to solve this optimization problem from geometric interpretation. Recall Equation (5), the $l_2$-norm can be interpreted as a radius constraint from the geometric perspective. Additionally, both the objective function and the cost

function are linear, indicating that the optimal solution lies on the boundary of the feasible region. By disregarding the element-wise bounds in Equation (5), we can consider the optimization problem as finding a optimal angle $\theta'$ on the $A\text{-}A_c$ plane, in accordance with Theorem 3.4. The optimal solution can be expressed as $\overline{\mathbf{v}} = 2N\delta'(\cos\theta'\tilde{\mathbf{A}}_c + \sin\theta'\tilde{\mathbf{A}})$, where $\tilde{\mathbf{A}}$ and $\tilde{\mathbf{A}}_c$ are the orthogonal unit vectors of $\mathbf{A}$ and $\mathbf{A_c}$ respectively. Considering Assumption 3.5, we proposed a iterative heuristic algorithm to solve Equation (5) by firstly calculating the optimal angle $\theta'$ regardless the element-wise bound and obtain a initial solution $\overline{\mathbf{v}}$, then clip $\overline{\mathbf{v}}$ according to the element-wise bound and mask the clipped value, and iteratively update the rest unmasked elements according to aforementioned steps until all elements in $\overline{\mathbf{v}}$ are satisfy the element-wise bound. The detailed steps are outlined in Appendix C. For the recovery update in Section 3.2.2, the same algorithm can be used to find the angle that satisfy $\overline{\mathbf{v}} \cdot \mathbf{A}_c = Nd'$ or $\overline{\mathbf{v}} \cdot \mathbf{A} = 0$.

**Theorem 3.4.** *Given a feasible optimization problem of the form:*

$$\underset{\overline{\mathbf{v}}}{\text{maximize}} \quad \overline{\mathbf{v}} \cdot \mathbf{A}$$
$$\text{s.t.} \quad \overline{\mathbf{v}} \cdot \mathbf{A}_c \leq D, \ \|\overline{\mathbf{v}}\|_2 \leq 2N\delta'$$
$$\mathbb{E}(\overline{\mathbf{v}}) = \mathbb{E}(\mathbf{A}) = \mathbb{E}(\mathbf{A}_c) = 0$$

*where $\overline{\mathbf{v}}$, $\mathbf{A}$, and $\mathbf{A}_c$ are $N$-dimensional vectors, then the optimal solution $\overline{\mathbf{v}}$ will lie in the $A\text{-}A_c$ plane determined by $\mathbf{A}_c$ and $\mathbf{A}$.*

**Assumption 3.5.** If the optimization problem in Theorem 3.4 has a optimal solution $\overline{\mathbf{v}}_{\text{opt}} = [\overline{v}_1, \overline{v}_2, \ldots]$, and the same problem with element-wise lower bound constraint $b$ has a optimal solution $\overline{\mathbf{v}}'_{\text{opt}} = [\overline{v}'_1, \overline{v}'_2, \ldots]$, then $\overline{v}'_t = b$ where $\overline{v}_t \leq b$.

*Remark* 3.6. By utilizing the proposed heuristic algorithm, the optimal solution to Equation (5) can be obtained in just a few iterations. The time complexity of each iteration is $O(n)$, where $n$ represents the number of unmasked elements. As a result, the computational complexity is significantly reduced compared to conventional convex optimization methods.

## 3.4 M-Step

After determining the optimal feasible posterior distribution $q$ to maximize the upper bound of ELBO, an M-step is implemented to maximize ELBO by updating policy parameters $\theta$ in a supervised learning manner. Recall the definition of ELBO in Equation (1) in Section 3.1, by dropping the part that independent from $\theta$, we will obtain following optimization problem

$$\underset{\theta}{\text{maximize}} \ -\alpha D_{\text{KL}}(q \parallel \pi_\theta) + \log p(\theta). \tag{7}$$

Note that if we assume $p(\theta)$ is a Gaussian distribution, then $\log p(\theta)$ can be converted into $D_{\text{KL}}(\pi \parallel \pi_\theta)$ (see Appendix B for details). Using the same trick in Section 3.2.1 to convert soft KL constraint to hard KL constraint, the supervised learning problem in M-step can be expressed as

$$\underset{\theta}{\text{minimize}} \ D_{\text{KL}}(q \parallel \pi_\theta)$$
$$\text{s.t.} \quad D_{\text{KL}}(\pi_\theta \parallel \pi) \leq \delta, \tag{8}$$

Note that $D_{\text{KL}}(\pi_\theta \parallel \pi)$ is chosen to lower than $\delta$ so that the current policy $\pi$ can be reached during the E-step in next update iteration to achieve robust update.

For Equation (7), it is a common practice for researchers to directly minimize the KL divergence, like CVPO (Liu et al., 2022) and MPO (Abdolmaleki et al., 2018). However, recall Equation (6), it is evident that the value of surrogate reward and cost are deeply connected to the projection of $\mathbf{v}$ onto the $A\text{-}A_c$ plane, while KL divergence can hardly reflect this kind of relationship between $\mathbf{v}$ and surrogate value. Consequently, Consequently, we choose to replace the original KL objective function with the $l^2$-norm $\mathbb{E}\left[\|v - p_{\pi_\theta}/p_\pi\|_2\right]$, where $v$ is the optimal probability ratio obtained in E-step and $p_{\pi_\theta}/p_\pi$ is the probability ratio under policy parameter $\theta$. With this replacement, the optimization problem can be treated as a fixed-target tracking control problem. This perspective enables us to plan tracking trajectories that can consistently satisfy the cost constraint, enhancing the ability to maintain cost satisfaction throughout the learning process. The optimization problem after replacement can be rewritten as

$$\underset{\theta}{\text{minimize}} \ \mathbb{E}\left[\|v - \frac{p_{\pi_\theta}}{p_\pi}\|_2\right] \qquad \text{s.t.} \quad D_{\text{KL}}(\pi_\theta \parallel \pi) \leq \delta, \tag{9}$$

To ensure the tracking trajectories can satisfy cost constraint at nearly all locations, we calculated the several recovery $\overline{\mathbf{v}'}$ under different $\delta''$ and guide $\frac{p_{\pi_\theta}}{p_\pi}$ to different $\overline{\mathbf{v}}$ according to the $l_2$-norm of $\frac{p_{\pi_\theta}}{p_\pi}$, so that even $\|\frac{p_{\pi_\theta}}{p_\pi}\|_2$ is much smaller than $2N\delta'$, the new policy can still satisfy the cost constraint. Moreover, inspired by the proportional navigation (Yanushevsky, 2018), we also modify the recovery update gradient from $(v - \frac{p_{\pi_\theta}}{p_\pi})\frac{\partial \pi_\theta}{\partial \theta}$ to $((\beta(v - \frac{p_{\pi_\theta}}{p_\pi}) + (1 - \beta)\mathbf{A}'_c)\frac{\partial \pi_\theta}{\partial \theta}$ to reduce the cost during the tracking, where $\mathbf{A}'_c$ is the projection of $v - \frac{p_{\pi_\theta}}{p_\pi}$ on cost advantage vector $\mathbf{A}_c$. In according with Theorem 3.7, the lower-bound clipping mechanism similar with PPO is applied on updating $\frac{p_{\pi_\theta}}{p_\pi}$ in M-step to satisfy the forward KL constraint (see Appendix C for details).

**Theorem 3.7.** *For a probability ratio vector $\overline{\mathbf{v}}$, if the variance of $\overline{\mathbf{v}}$ is constant, then the upper bound of the approximated forward KL divergence $D_{\mathrm{KL}}(\pi_\theta \parallel \pi)$, will decrease as the element-wise lower bound of $\overline{\mathbf{v}}$ increase.*

Apart from E-step and M-step introduced in Section 3.2 and Section 3.4, our method shares the same Generalized Advantage Estimator (GAE) technique (Schulman et al., 2015b) with PPO in calculating the advantage value $A$ and $A_c$. The main steps of CPPO are summarized in Appendix C.

# 4 Experiment

In this section, Safety Gym (Ray et al., 2019) benchmark environments and Circle environment (Achiam et al., 2017) are used to verify and evaluate the performance of the proposed method. Five test scenarios, namely CarPush, PointGoal, PointPush, PointCircle, and AntCircle are evaluated. The detailed information about the test scenarios can be seen in Appendix D. Three algorithms are chosen as the benchmarks to compare the learning curves and the constraint satisfaction: CPO (Achiam et al., 2017), PPO-Lagrangian method (simplified as PPO_lag), and TRPO-Lagrangian method (simplified as TRPO_lag) (Ray et al., 2019). CPO is chosen as the representative of the feasible region method. PPO_lag and TRPO_lag are treated as the application of the primal-dual method in first-order optimization and second-order optimization. TRPO and PPO are also used in this section as unconstrained performance references. For a fair comparison, all of the algorithms use the same policy network and critic network. The detail of the hyperparameter setting is listed in Appendix E.

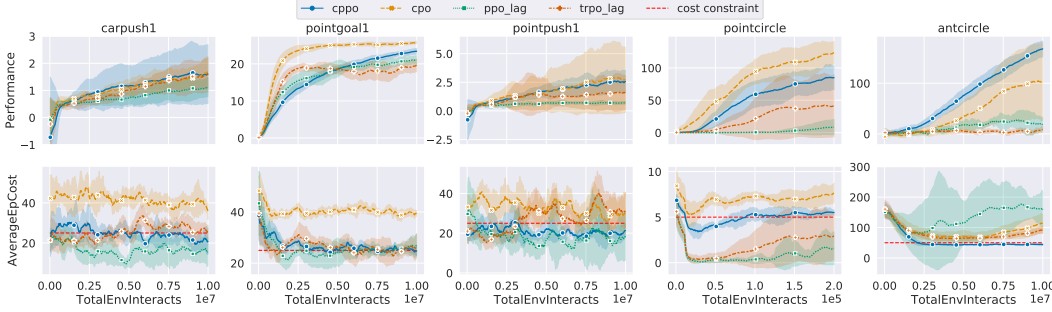

Figure 3: The learning curves for comparison, CPPO is the method proposed in this paper.

**Performance and Constraint Satisfaction:** Figure 3 compares the learning curves of the proposed method and other benchmark algorithms in terms of the episodic return and the episodic cost. The first row records the undiscounted episodic return for performance comparison, and the second row is the learning curves of the episodic cost for constraint satisfaction analysis, where the red dashed line indicates the cost constraint. The learning curves for the Push and Goal environments are averaged over 6 random seeds, while those for the Circle environments are averaged over 4 random seeds. The curve itself represents the mean value, and the shadow indicates the standard deviation. In terms of performance comparison, it was observed that CPO can achieve the highest reward return in PointGoal and PointCircle. The proposed CPPO method, on the other hand, achieves similar or even higher reward return in the remaining test scenarios. However, when considering constraint satisfaction, CPO fails to satisfy the constraint in all four tasks due to approximation errors, as

previously reported in Ray et al. (2019). In contrast, CPPO successfully **satisfies the constraint** in all five environments, showing the effectiveness of the proposed **recovery update** . Referring to the learning curves in Circle scenarios, it can be seen that the primal-dual based CRL methods, i.e., PPO_lag and TRPO_lag, suffer from the slow and unstable update of the dual variable, causing the conservative performance in PointCircle and slow cost satisfaction in AntCircle. On the other hand, CPPO can achieves a faster learning speed in Circle environment by **eliminating the need for the dual variable**. Overall, the experimental results demonstrate the effectiveness of CPPO in solving the CRL problem.

**Ablation Study:** An ablation study was conducted to investigate the impact of the recovery update in CPPO. Figure 4 presents the reward performance and cost satisfaction of CPPO with and without the recovery update in the PointCircle environment. The results indicate that without the recovery update, CPPO achieves higher reward performance; however, the cost reaches 15, which significantly violates the cost constraint. In contrast, when the recovery update is applied, CPPO successfully satisfies the constraint, thereby demonstrating the importance of the recovery update in ensuring constraint satisfaction.

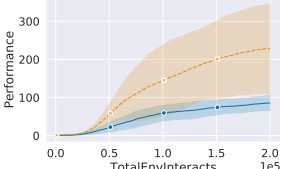 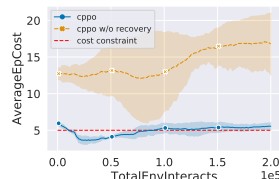

Figure 4: The comparison between CPPO with and without recovery update in PointCircle.

# 5 Limitations and Boarder Impact

Although our proposed method has shown its ability in test scenarios, there still exist some limitations. Firstly, CPPO method is an on-policy constrained RL, which suffers from lower sampling efficiency compared to other off-policy algorithms, potentially limiting its applicability in real-world scenarios. Additionally, the convergence of our method is not yet proven. However, we believe that our work will offer researchers a new EM perspective for using PPO-like algorithms to solve the problem of constrained RL, thereby leading to the development of more efficient and stable constrained RL algorithms.

# 6 Conclusion

In this paper, we have introduced a novel first-order Constrained Reinforcement Learning (CRL) method called CPPO. Our approach avoids the use of the primal-dual framework and instead treats the CRL problem as a probabilistic inference problem. By utilizing the Expectation-Maximization (EM) framework, we address the CRL problem through two key steps: the E-step, which focuses on deriving a theoretically optimal policy distribution, and the M-step, which aims to minimize the difference between the current policy and the optimal policy. Through the non-parametric representation of the policy using probability ratios, we convert the CRL problem into a convex optimization problem with a clear geometric interpretation. As a result, we propose an iterative heuristic algorithm that efficiently solves this optimization problem without relying on the dual variable. Furthermore, we introduce a recovery update strategy to handle approximation errors in cost evaluation and ensure constraint satisfaction when the current policy is infeasible. This strategy mitigates the impact of approximation errors and strengthens the capability of our method to satisfy constraints. Notably, our proposed method does not require second-order optimization techniques or the use of the primal-dual framework, which simplifies the optimization process. Empirical experiments have been conducted to validate the effectiveness of our proposed method. The results demonstrate that our approach achieves comparable or even superior performance compared to other baseline methods. This showcases the advantages of our method in terms of simplicity, efficiency, and performance in the field of Constrained Reinforcement Learning.

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
