# OpenReview forum: "Constrained Proximal Policy Optimization"
_NeurIPS.cc/2023/Conference — Submitted to NeurIPS 2023_

### Official Review · Reviewer_ChuP · 2023-06-26

**Soundness:** 3 good
**Presentation:** 2 fair
**Contribution:** 3 good
**Rating:** 3
**Confidence:** 4

**Summary:**

This paper introduces the constrained version of PPO that is devised for solving constrained MDP problems in the discounted reward setting with discounted cost constraints.


**Strengths:**

The paper presents the constrained PPO. It also gives some analytical results and presents a heuristic algorithm that is seen to perform well numerically.

**Weaknesses:**

1. Inadequate literature survey. The authors should broaden the scope of their survey - in fact constrained actor critic was proposed in the paper: "V.S.Borkar, An actor-critic algorithm for constrained Markov decision processes, Systems and Control Letters, 54(3):207-213, 2005'', for the full state case and "S.Bhatnagar, An actor–critic algorithm with function approximation for discounted cost constrained Markov decision processes, Systems and Control Letters, 59(12): 760-766, 2010, in the function approximation setting.

2. The authors point to the fact that they have not provided a convergence analysis for their algorithm even though other works in the literature such as the ones listed above do possess an asymptotic analysis of convergence.

3. The results given in the text carry imprecise statements, for instance, Prop. 3.1 says "if there are a sufficient number of sampled v, then E[v]=1 and E[vlog v] <= var(v-1). This has to be made precise. Does it  mean that in the limit that the number of samples goes to infinity, the statement is valid? Then, if so, the question will be what will be the form of the statement in terms of the number of samples N.

4. The recovery update problem in Sec 3.2 is not clear - how it has been arrived at?

5. Subsequently a heuristic procedure has been proposed and the authors say that an optimal solution to (5) is provably obtained, But if it is provably obtained, how is the procedure a heuristic procedure?

6. After (4), it is said that (4) can be directly solved through existing convex optimization techniques. This is not clear since one needs A(s,a), A_c(s,a) etc. to be known in order to solve it directly. But that is not known and needs to be estimated. So not clear what the authors mean?
****
7. On further reading of the paper and supplementary material post the response of the authors, I feel the technical results are imprecise and flawed. For instance, there is no way to verify Assumption 3.5. Moreover, the statement of Proposition 3.1 seems to suggest that it depends on a "sufficient number of sampled v" but gives a bound on the true expectation. Why should the true expectation depend on the number of samples of v? Also, when you say "sufficient number of sampled v", what does it mean? How many v is a sufficient number?


**Questions:**

1. On page 2, is it \pi(a|s) instead of \pi(s|a)?
2. Beginning of page 3: definitions of Q(s_t,a_t), V(s_t), Q_c(s_t,a_t) and V_c(s_t) are confusing. This is because you have a summation of t from 0 to \infty, and then you condition on s_0=s_t, a_0=a_t etc. What does it mean? It is too confusing.
3. After (1), how is d^\pi defined?
4. In (3), is there a relationship between d and \delta? Can one be found from the other?


**Limitations:**

The major limitation is in terms of a lack of credible analysis. Even the theoretical results presented are not precise, so nothing much can be said about the algorithm. In addition there are several typos and grammatical errors throughout the paper.

---

> ### Author Rebuttal · Authors · 2023-08-08
>
> **We would like to express our sincere gratitude to you for valuable and constructive comments. Our responses are as follows.**
>
> Some comments from reviewers are not displayed in full but replaced by ... to save space.
>
> **W1**: Inadequate literature survey, ... , in the function approximation setting.
>
> **R1**: Thank you for your comment. We have carefully reviewed the suggested papers and incorporate them into our revised paper's citation list.
>
> **W2**: The authors point to the fact that ... listed above do possess an asymptotic analysis of convergence.
>
> **R2**: Thank you for your comment. The convergence analyses presented in the aforementioned papers rely on treating the Lagrangian multiplier as a **constant** and demonstrate the convergence of the V-value, as defined under this constant multiplier, using the contraction mapping theorem. It is important to note that these methods are **specifically tailored for the Lagrangian-based approach**.
>
> However, the feasible region method like CPO, CVPO, and the proposed CPPO method operates without the necessity of a global Lagrangian multiplier for optimization. Consequently, the convergence analyses put forth in the mentioned papers cannot be directly applied to prove the convergence of our approach.
>
> **W3**: The results given in the text carry imprecise statements, ..., in terms of the number of samples N.
>
> **R3**: Thank you for your comment. It is evident that $v$, being the probability ratios, has an expected value of $1$. This implies that if we were to sample a large number of instances of $v$, the mean value of these samples would equal 1. Consequently, we believe that the statements presented in Proposition 3.1 are precise.
>
> However, in practice, we can observe that even when hundreds of $v$ samples are considered, the mean value of $v$ can still be very close to 1. This observation is particularly evident in the PPO method, where one can calculate the mean value of $v$ during each optimization step. As an on-policy RL methods, it is common to obtain thousands or even tens of thousands of sampled $v$ in a single rollout, thereby affirming the validity of Proposition 3.1 for the proposed method.
>
> **W4**: The recovery update problem in Sec 3.2 is not clear - how it has been arrived at?
>
> **R4**: Thank you for your comment. As mentioned in Sec. 3.2.2,  the purpose of the recovery update is to bring the current policy back from the infeasible region. This involves reducing the episode cost of the current policy by finding an alternative policy that *minimizes costs while either preserving or minimizing any reduction in the overall reward return*. In other words, we seek to minimize the episode cost return without negatively impacting the episode reward return. Consequently, obtaining the three recovery cases shown in Fig. 1 is a straightforward process.
>
> **W5**: Subsequently a heuristic procedure has been proposed, ... , how is the procedure a heuristic procedure?
>
> **R5**: Thank you for your comment. We acknowledge that the paper does not explicitly state that we have *proved* the optimal solution is attained. In Section 3.3, we refer to the algorithm as a *heuristic* procedure because it relies on **Assumption 3.5**, as stated in **Line 258**. If this assumption is validated, the algorithm will be demonstrated to yield the optimal solution. Presently, the assumption remains unproven; however, we firmly believe in its correctness based on our verification of the solution against the results obtained from Matlab's *fmincon* function.
>
> **W6**: After (4), ... So not clear what the authors mean?
>
> **R6**: Thank you for your comment. Before proceeding with the optimization in equation (4), we performed an estimation of both $A(s,a)$ and $A_c(s,a)$ using GAE technique[1]. This approach is widely adopted in various RL algorithms, such as PPO and MPO. It's important to note that we obtained the advantage-value/q-value **prior** to initiating the optimization process.
>
> **Q1**: On page 2, is it \pi(a|s) instead of \pi(s|a)?
>
> **A1**: Thank you for your comment, we have reviewed and revised the typo in the paper according to your comments.
>
> **Q2**: Beginning of page 3: ... It is too confusing.
>
> **A2**: Thank you for your comment. We acknowledge that the definitions of $Q$, $V$, $Q_c$, and $V_c$ in our paper are consistent with those used in other works within the CRL field. Nevertheless, we understand the importance of clarity and will make revisions to enhance the definitions for better understanding.
>
> **Q3**: After (1), how is d^\pi defined?
>
> **A3**: Thank you for your comment. As defined in **Line 162**, $d^{\pi}$ represents the state distribution under the current policy $\pi$. In practical implementations, this state distribution is implicitly represented by the states of the trajectories sampled from policy $\pi$, following the same representation method used in PPO and TRPO.
>
> **Q4**: In (3), is there a relationship between d and \delta? Can one be found from the other?
>
> **A4**: Thank you for your comment, the definition of $d$ can be found in **Line 165**, where it represents the cost constraint. On the other hand, $\delta$ corresponds to the reverse KL constraint, as defined in **Line 177**. It is essential to note that these two parameters are independent of each other, each serving a distinct purpose in our study.
>
> **Lastly, we would like to express our gratitude for your patience in reviewing our response, and for your invaluable assistance in enhancing our paper thus far! Please let us know if you have any further questions. We are actively available!**
>
> \[1\] Schulman, John, et al. \"High-dimensional continuous control using generalized advantage estimation.\" arXiv preprint arXiv:1506.02438 (2015).

---

> > ### Comment · Reviewer_ChuP · 2023-08-15
> > **response to rebuttal for referee ChuP**
> >
> > R2: No, the Lagrange multiplier in the mentioned references is not constant but those papers have an update rule and they prove the convergence of even the Lagrange multiplier. Please take a closer look.
> >
> > I have read through all the responses of the authors. While I appreciate the responses provided by the authors, I am convinced the paper lacks concrete analysis. In particular, Assumption 3.5 makes no sense.  In Proposition 3.1, the statement reads that if there are a sufficient number of sampled v, then E[v]=1 and E[vlog v] \leq var(v-1). Why should such a result depend on the number of samples of v when E[.] and var(.) are meant to be the true expectation and variance respectively? These are the imprecisions I am pointing to in the review. Since the results are technically flawed, I am reducing my rating of the paper to 3. I have also updated my review.

---

> > > ### Author Response · Authors · 2023-08-15
> > >
> > > Thank you for the further comments!
> > >
> > > For R2, when they prove the convergence of the **value function**, they treat the Lagrange multiplier as a constant. This assumption doesn't affect their method's Lagrange multiplier updates. However, it is crucial to highlight that this approach is **not suitable for proving the convergence of the trust region method**. This is because the trust region method doesn't have a global Lagrange multiplier.
> > >
> > > For Assumption 3.5, we **partly confirmed this hypothesis by testing it using a Matlab function**. If you have doubts about this assumption, it could be helpful to find counterexamples to support your point. Also, note that E[] and var() represent the average and variance of  **sampled** probability ratios. These are different from the actual expected values and variances of probability ratios, which don't change based on the number of samples used.
> > >
> > > **Lastly, we would like to express our gratitude for your patience in reviewing our response.**

---

### Official Review · Reviewer_qYFp · 2023-07-03

**Soundness:** 2 fair
**Presentation:** 1 poor
**Contribution:** 2 fair
**Rating:** 4
**Confidence:** 2

**Summary:**

The paper introduces Constrained Proximal Policy Optimization (CPPO) for Constrained Reinforcement Learning (CRL). The CPPO method is designed to overcome the limitations of existing methods by offering a first-order feasible region method that doesn't require dual variables or second-order optimization, and it is an incremental extension of the  CVPO algorithm (Constrained Variational Policy Optimization for Safe Reinforcement Learning). The improvement seems incremental, improving the computational efficiency of CVPO. The method is evaluated in different environment,  comparable or even superior performance to other baseline methods.  . However, the paper does not provide  a direct comparison between CPPO and CVPO.

**Strengths:**

- The authors propose a new first-order method for constrained RL. The method seems to be designed to be simple, and overcome  limitations of existing methods such as CPO and CVPO (e.g., the authors do not require the usage of a dual variable).

- The proposed method demonstrates comparable or even superior performance compared to other baseline methods.

**Weaknesses:**

- Clarity: the paper could benefit from improvements in clarity and readability. The presentation of their ideas is somewhat dense and could be difficult for readers to follow. A concise description of the algorithm is missing.

- Novelty and results: The method builds directly upon CVPO. The authors need to clearly highlight this fact when introducing their method, and clearly explain what are the differences. As of now, the changes seem incremental, and therefore the paper seems to lack in novelty. Furthermore, the paper does not provide  a direct comparison between CPPO and CVPO, especially in terms of computational complexity

- No theoretical results are provided, therefore I'd have expected more extensive numerical results.

- Typos and notation: there are several typos and errors in notation throughout the paper.

**Questions:**

See above

**Limitations:**

Authors briefly discuss limitations and broader impact.

---

> ### Author Rebuttal · Authors · 2023-08-08
>
> **We would like to express our sincere gratitude to you for valuable and constructive comments. Our responses are as follows.**
>
> **W1**: Clarity: the paper could benefit from improvements in clarity and readability. The presentation of their ideas is somewhat dense and could be difficult for readers to follow. A concise description of the algorithm is missing.
>
> **R1**: Thank you for your comment, we have reviewed and revised the paper according to your comments.
>
> **W2**: Novelty and results: The method builds directly upon CVPO. The authors need to clearly highlight this fact when introducing their method, and clearly explain what are the differences. As of now, the changes seem incremental, and therefore the paper seems to lack in novelty. Furthermore, the paper does not provide a direct comparison between CPPO and CVPO, especially in terms of computational complexity
>
> **R2**: Thanks for your comment. We believe the proposed CPPO method is an extension of PPO in constrained RL field, rather than a straightforward on-policy CVPO approach. The only thing these two methods in common is that they share the same idea of finding an optimal policy within a trust region and progressively pushing the current policy towards the optimal one in an EM manner. Similar ideas have been successfully applied in CPO, MPO, and V-MPO as well.
>
> The most important contribution of our work is using the **probability ratio** instead of **probability density** to represent the optimal distribution. Previous MPO-based algorithms (MPO, CVPO, V-MPO) endeavour to directly derive the probability density $\psi$ according to $\int\psi(a|s)da=1$. However, note that this formulation does not inherently yield that $\sum\psi(a|s)=1$, **which is a error persists across all three aforementioned algorithms**, leading to the **incorrect normalization** of $\psi^*$. In contrast, our work addresses this issue by employing the probability ratio $v$, which allows for a more straightforward calculation of the distribution with ensuring $E(v)=1$.
>
> Furthermore, in contrast to the KL divergence, the utilization of the $l_2$-norm ($\chi^2$ divergence) during the E-step offers a distinct **geometric interpretation** of the feasible region. This perspective facilitates the formulation of the **recovery update method**, which effectively minimizes costs while maintaining rewards.  In M-step, the proposed CPPO algorithm conduct a policy update process akin to that of PPO, thereby obviating the necessity for additional hyperparameters as seen in other MPO-based approaches.
>
> For the computational complexity, based on our experiment, with the same number of sampled states,  CVPO requires approximately 5s for one epoch update, whereas CPPO achieves the same update in less than 2s.
>
> **W3**:No theoretical results are provided, therefore I'd have expected more extensive numerical results.
>
> **R3**: Thank you for your comment. We acknowledge that due to limited resources, we could only conduct tests on a restricted number of benchmark environments. In our future work, we will make a effort to include additional numerical experiments.
>
> **W4**: Typos and notation: there are several typos and errors in notation throughout the paper.
>
> **R4**:  Thanks for your comment. We have reviewed the entire paper and make necessary revisions to address typos and notation errors in the paper according to your comments.
>
> **Lastly, we would like to express our gratitude for your patience in reviewing our response, and for your invaluable assistance in enhancing our paper thus far! Please let us know if you have any further questions. We are actively available!**

---

> > ### Comment · Reviewer_qYFp · 2023-08-19
> >
> > I sincerely thank the authors for taking the time to answer my concerns. Although I acknowledge the contributions, an effort needs to be made to improve the readability of the paper (especially to better emphasize the difference w.r.t. CVPO) and the technical analysis. I'm grateful for the authors' dedication to responding to the reviewers, and I'm eager to observe the enhancements in future versions.

---

> > > ### Author Response · Authors · 2023-08-19
> > >
> > > Thank you for the further comments! We will continue to revise and polish our work according to your advice. Your feedback is truly valuable to us!

---

### Official Review · Reviewer_XWVs · 2023-07-04

**Soundness:** 3 good
**Presentation:** 3 good
**Contribution:** 3 good
**Rating:** 5
**Confidence:** 4

**Summary:**

This paper proposes a novel first-order feasible method, CPPO, for efficient constrained reinforcement learning. The proposed approach integrates the Expectation-Maximization (EM) framework to solve the policy optimization problem by treating the CRL as the probabilistic inference. In the E-step, CPPO calculates the optimal policy distribution within the feasible region. In the M-step, CCPO conducts a first-order update for policy optimization. The authors also propose an iterative heuristic algorithm from a geometric perspective to efficiently solve the E-step and a recovery update strategy to improve constraint satisfaction performance. They evaluate their algorithm in several benchmark environments. The reported results show comparable performance over other baselines in complex environments.


**Strengths:**

(1) The proposed algorithm converts the CRL problem into a convex optimization problem with a clear geometric interpretation which mitigates the impact of approximation errors and strengthens the capability of the proposed method to satisfy constraints.

(2) Since the proposed method does not require second-order optimization techniques or the use of the primal-dual framework, the policy optimization process is largely simplified.


**Weaknesses:**

(1) I suggest further polishing and improving the mathematical formulation and notations to make them more rigorous. For example, the objectives and constraints in (5) (6) are represented by the dot product of two vectors. In my understanding, it should be easier to understand with a transpose on the top of the first vector.

(2) The policy update strategy looks overly conservative. Figure 1 shows that only in case 3, the policy can be updated toward seeking a higher reward. Does this strategy result in over-conservativeness? I am also wondering in cases 1 and 3 why the solution is not at the feasible boundary (that is to find the feasible distribution to maximize A).

(3) It is very impressive that CPPO can work well in the AntCircle and Push tasks. However, the experiment lacks sufficient baselines and environments for comparison. For example, the recovery update in Section 3.2.2 is similar to the idea in Yang et. al [1, 2], where an additional projection step is introduced to recover the safe policy, so I am wondering how does the author compare CPPO against these methods, both in theory and empirically. In addition, the results only present a subset of tasks in SafetyGym, so I wonder how the algorithm performs in other tasks.

[1] Yang, Tsung-Yen, et al. "Projection-based constrained policy optimization." arXiv preprint arXiv:2010.03152 (2020).

[2] Yang, Tsung-Yen, et al. "Accelerating safe reinforcement learning with constraint-mismatched baseline policies." International Conference on Machine Learning. PMLR, 2021.

**Questions:**

(1) This work seems like an on-policy version of CVPO, so I am wondering how does the proposed approach compare against CVPO in the experiments?
(2) Could the authors provide more details about baseline implementations, such as the PPO-Lag and TRPO-lag? I am not able to find them in the supplementary material.


**Limitations:**

The authors adequately addressed the limitations.

---

> ### Author Rebuttal · Authors · 2023-08-08
>
> **We would like to express our sincere gratitude to you for valuable and constructive comments. Our responses are as follows.**
>
> **W1**: I suggest further polishing and improving the mathematical formulation and notations to make them more rigorous. For example, the objectives and constraints in (5) (6) are represented by the dot product of two vectors. In my understanding, it should be easier to understand with a transpose on the top of the first vector.
>
> **R1**: Thank you for your comment, we have reviewed and revised the paper according to your comments.
>
> **W2**: The policy update strategy looks overly conservative. Figure 1 shows that only in case 3, the policy can be updated toward seeking a higher reward. Does this strategy result in over-conservativeness? I am also wondering in cases 1 and 3 why the solution is not at the feasible boundary (that is to find the feasible distribution to maximize A).
>
> **R2**: Thank you for your comment. The recovery update strategy is selectively employed when the current policy **violates the cost constraint**. It is essential to consider that the distribution obtained from the E-step might not adhere strictly to a Gaussian distribution, potentially hindering the actor policy's ability to reach the optimal policy. Therefore, we try to introduce some over-conservativeness to guide the current policy back to the feasible region. As a result, in cases 1&3, our proposed solution lies on the boundary of the trust region, rather than solely on the boundary of the feasible region."
>
> **W3**:It is very impressive that CPPO can work well in the AntCircle and Push tasks. However, the experiment lacks sufficient baselines and environments for comparison. For example, the recovery update in Section 3.2.2 is similar to the idea in Yang et. al [1, 2], where an additional projection step is introduced to recover the safe policy, so I am wondering how does the author compare CPPO against these methods, both in theory and empirically. In addition, the results only present a subset of tasks in SafetyGym, so I wonder how the algorithm performs in other tasks.
>
> **R3**: Thank you for your comment. We believe PCPO and SPACE are effective representations for efficiently solving CRL problems through the use of projection methods. The incorporation of projection allows for a substantial reduction in the number of constraint violations, enhancing their overall performance. However, it's important to acknowledge that these projection methods still rely on first/second-order optimization to estimate the cost, which may lead to cost violations in complex environments, as exemplified by CPO's performance in Safety Gym.
>
> In contrast, the recovery update strategy is designed to maximize cost reduction while preserving the reward during the recovery phase. This approach helps mitigate the side effects of approximation errors, making it a promising alternative to traditional projection methods in theory.
>
> Due to the limited computing resources, we were only able to test a few environments. In future work, we plan to expand the scope and incorporate additional test scenarios to further validate and enhance the findings of our research.
>
> **Q1**: This work seems like an on-policy version of CVPO, so I am wondering how does the proposed approach compare against CVPO in the experiments?
>
> **A1**: Thank you for your comment. Considering that CVPO is an off-policy algorithm and due to the limited computing resources within our team, we have not included this algorithm as a benchmark in our current study. Nevertheless, we acknowledge its significance and will include it for comparison in our future work.
>
> **Q2**: Could the authors provide more details about baseline implementations, such as the PPO-Lag and TRPO-lag? I am not able to find them in the supplementary material.
>
> **A2**: Thank you for your comment. The baseline implementations in our paper are built upon the code provided by Safety Gym [1]. The code can be accessed at <https://github.com/openai/safety-starter-agents>
>
> **Lastly, we would like to express our gratitude for your patience in reviewing our response, and for your invaluable assistance in enhancing our paper thus far! Please let us know if you have any further questions. We are actively available!**
>
> \[1\] Ray, Alex, Joshua Achiam, and Dario Amodei. \"Benchmarking safe exploration in deep reinforcement learning.\" arXiv preprint arXiv:1910.01708 7.1 (2019): 2.

---

> > ### Comment · Reviewer_XWVs · 2023-08-13
> >
> > Thank you for your response. Having reviewed the other reviews and the corresponding rebuttals, I have decided to retain my original score. While I acknowledge the contributions of the paper, I believe that conducting more comprehensive comparison experiments and providing deeper theoretical analysis could significantly enhance the quality of the work in future versions. I appreciate the authors' efforts in addressing the feedback and look forward to seeing potential improvements in subsequent iterations.

---

> > > ### Author Response · Authors · 2023-08-13
> > >
> > > Thank you for the further comments! We will continue to revise and polish our work according to your advice. Your feedback is truly valuable to us!

---

### Official Review · Reviewer_DAoZ · 2023-07-26

**Soundness:** 3 good
**Presentation:** 2 fair
**Contribution:** 3 good
**Rating:** 5
**Confidence:** 3

**Summary:**

This paper proposes a novel CPPO method to solve the contained RL (CRL) problem. Specifically, CPPO leverages the probabilistic inference and converts the CRL problem formulation based on the probabilistic ratio, resulting in the first-order optimization solution. CPPO also develops the recovery update method to safely optimize the policy when there are inaccurate cost evaluations and infeasible solutions.
The resulting EM-based framework of CPPO shows its effectiveness across various safety gym scenarios compared to baselines.

**Strengths:**

1. CPPO solves the constrained RL based on first-order and does not use dual variables and second-order optimization, resulting in a simpler, intuitive (i.e., geometric perspective), and computationally efficient method.
2. The recovery update method is developed thanks to CPPO's first-order optimization/geometric perspectives and shows its effectiveness (as shown in Figure 4).


**Weaknesses:**

1. As stated in Section 3.2.1, CPPO builds on CVPO and has two main differences (using advantage instead of q and using the probability ratio instead of directly calculating q). While  Some readers may understand these as a limited novelty. Possibly adding comparisons against CVPO in the evaluation section can convey the importance of these differences better.
2. While I agree that CPPO reduces computational complexity, there are no empirical results in the evaluation section. Possibly, adding computation time results in the evaluation section can help.
3. I understand the benefits of converting the CRL problem into first-order optimization, but it is unclear what potential limitations/disadvantages the first-order optimization may have. Could there be an approximation error compared to second-order optimization? Related to this concern, could the authors clarify further why Assumption 3.5 is a fair assumption?


**Questions:**

1. Could the CPO baseline or other baselines also apply a similar recovery update strategy (i.e., cases 1-3 described in Section 3.2.2)?

**Limitations:**

The authors have adequately addressed the limitations.

---

> ### Author Rebuttal · Authors · 2023-08-08
>
> **We would like to express our sincere gratitude to you for valuable and constructive comments. Our responses are as follows.**
>
> **W1**: As stated in Section 3.2.1, CPPO builds on CVPO and has two main differences (using advantage instead of q and using the probability ratio instead of directly calculating q). While Some readers may understand these as a limited novelty. Possibly adding comparisons against CVPO in the evaluation section can convey the importance of these differences better.
>
> **R1**: Thanks for your comment. We believe the proposed CPPO method is an extension of PPO in constrained RL field, rather than a straightforward on-policy CVPO approach. The only thing these two methods in common is that they share the same idea of finding an optimal policy within a trust region and progressively pushing the current policy towards the optimal one in an EM manner.
>
> The most important contribution of our work is using the **probability ratio** instead of **probability density** to represent the optimal distribution. Previous MPO-based algorithms (MPO, CVPO, V-MPO) endeavour to directly derive the probability density $\psi$ according to $\int\psi(a|s)da=1$. However, note that this formulation does not inherently yield that $\sum\psi(a|s)=1$, **which is a error persists across all three aforementioned algorithms**, leading to the **incorrect normalization** of $\psi^*$. In contrast, our work addresses this issue by employing the probability ratio $v$, which allows for a more straightforward calculation of the distribution with ensuring $E(v)=1$.
>
> Furthermore, in contrast to the KL divergence, the utilization of the $l_2$-norm ($\chi^2$ divergence) during the E-step offers a distinct **geometric interpretation** of the feasible region. This perspective facilitates the formulation of the **recovery update method**, which effectively minimizes costs while maintaining rewards.  In M-step, the proposed CPPO algorithm conduct a policy update process akin to that of PPO, thereby obviating the necessity for additional hyperparameters as seen in other MPO-based approaches.
>
> Considering that CVPO is an off-policy algorithm, we have not included it as a benchmark in our current study. Nonetheless, we acknowledge its significance and plan to incorporate it for comparison in our future work.
>
> **W2**: While I agree that CPPO reduces computational complexity, there are no empirical results in the evaluation section. Possibly, adding computation time results in the evaluation section can help.
>
> **R2**: Thank you for your comment. Based on our experiment, when using the same number of sampled states, the CVPO algorithm requires approximately 5 seconds for one epoch update, whereas CPPO only takes less than 2 seconds for one epoch update. In our future work, we plan to include further computation time comparisons to gain a more comprehensive understanding of the algorithms' performance.
>
> **W3**: I understand the benefits of converting the CRL problem into first-order optimization, but it is unclear what potential limitations/disadvantages the first-order optimization may have. Could there be an approximation error compared to second-order optimization? Related to this concern, could the authors clarify further why Assumption 3.5 is a fair assumption?
>
> **R3**: Thanks for your comment. You are correct in pointing out that first-order optimization can introduce errors, primarily due to the fact that the optimal distribution calculated in the E-step may not align perfectly with a Gaussian distribution. As a result, the actor policy may not precisely reach the optimal policy. This limitation also necessitates the implementation of a recovery update strategy.
>
> For Assumption 3.5, this assumption is based on the geometric intuition that the optimal solution of equation (5) will consistently lie on the boundary of the feasible region. To validate our hypothesis, we conducted a comparison between our solution and the results obtained using Matlab's *fmincon* function, affirming the accuracy of our approach. Thus, we firmly believe in the validity of this hypothesis.
>
> **Q1**: Could the CPO baseline or other baselines also apply a similar recovery update strategy (i.e., cases 1-3 described in Section 3.2.2)?
>
> **A1**: Thanks for your comment. The recovery update strategy is specifically tailored for the feasible region method, indicating its theoretical applicability to CPO. However, the practical implementation of this recovery process on CPO could be challenging. For the primal-dual approaches, they doesn't necessarily require this strategy, as its convergence relies on the convergence of the Lagrange multiplier.
>
> **Lastly, we would like to express our gratitude for your patience in reviewing our response, and for your invaluable assistance in enhancing our paper thus far! Please let us know if you have any further questions. We are actively available!**

---

> > ### Comment · Reviewer_DAoZ · 2023-08-14
> >
> > I appreciate the authors' clarifications. They answer my questions and concerns about this paper. After reading other reviewers' comments, I would like to retain my rating, which I am unsure about the acceptance. While I acknowledge the paper's technical contributions, the story of the paper may need to be entirely re-written to highlight the difference against CVPO (e.g., highlighting the issue of incorrect normalization and why this incorrect normalization is problematic in practice) as most of the reviewers are concerned about the novelty aspect. I also understand that directly comparing against CVPO is not straightforward, but having former comparisons (e.g., computation time, correct normalization) against the baseline is important. Thank you again for clarifying my questions.

---

> > > ### Author Response · Authors · 2023-08-15
> > >
> > > Thank you for the further comments! We will continue to revise and polish our work according to your advice. Your feedback is truly valuable to us!

---

### Official Review · Reviewer_8NjX · 2023-07-27

**Soundness:** 3 good
**Presentation:** 3 good
**Contribution:** 2 fair
**Rating:** 5
**Confidence:** 2

**Summary:**

This paper studies constrained reinforcement learning problems. It proposes a new EM-type algorithm and designs a heuristic version for practical use. The authors also conduct some numerical experiments to validate the performance of the algorithm.

**Strengths:**

1. The algorithm is first-order and thus computationally efficient in practice.
2. The numerical results look convincing.

**Weaknesses:**

1. The algorithm looks very similar to CVPO and only has some small modifications.
2. This paper does not have convincing theoretical analysis. For example, the authors claim that the heuristic algorithm in Section 3.3 will stop in just a few iterations (remark 3.6) but there are not any theoretical proofs. It would be better if the paper can provide some theoretical study, even only for simple cases.

**Questions:**

None.

**Limitations:**

None.

---

> ### Author Rebuttal · Authors · 2023-08-08
>
> **We would like to express our sincere gratitude to you for valuable and constructive comments, thanks for your support. Our responses are as follows.**
>
> **W1**: The algorithm looks very similar to CVPO and only has some small modifications.
>
> **R1**: Thanks for your comment. We believe the proposed CPPO method is an extension of PPO in constrained RL field, rather than a straightforward on-policy CVPO approach. The only thing these two methods in common is that they share the same idea of finding an optimal policy within a trust region and progressively pushing the current policy towards the optimal one in an EM manner. Similar ideas have been successfully applied in CPO, MPO, and V-MPO as well.
>
> The most important contribution of our work is using the **probability ratio** instead of **probability density** to represent the optimal distribution. Previous MPO-based algorithms (MPO, CVPO, V-MPO) endeavour to directly derive the probability density $\psi$ according to $\int\psi(a|s)da=1$. However, note that this formulation does not inherently yield that $\sum\psi(a|s)=1$, **which is a error persists across all three aforementioned algorithms**, leading to the **incorrect normalization** of $\psi^*$. In contrast, our work addresses this issue by employing the probability ratio $v$, which allows for a more straightforward calculation of the distribution with ensuring $E(v)=1$.
>
> Furthermore, in contrast to the KL divergence, the utilization of the $l_2$-norm ($\chi^2$ divergence) during the E-step offers a distinct **geometric interpretation** of the feasible region. This perspective facilitates the formulation of the **recovery update method**, which effectively minimizes costs while maintaining rewards.  In M-step, the proposed CPPO algorithm conduct a policy update process akin to that of PPO, thereby obviating the necessity for additional hyperparameters as seen in other MPO-based approaches.
>
> **W2**: This paper does not have convincing theoretical analysis. For example, the authors claim that the heuristic algorithm in Section 3.3 will stop in just a few iterations (remark 3.6) but there are not any theoretical proofs. It would be better if the paper can provide some theoretical study, even only for simple cases.
>
> **R2**: Thanks for your comment. The heuristic algorithm presented in Section 3.3 exhibits a two-step recursion process, effectively addressing the optimization problem in (5). When the solution encompasses a set of $N$ elements, the algorithm can be expressed as follows:
>
> * Step 1: We neglect the lower bound constraint in (5) and derive an optimal solution following Theorem 3.4. If this solution satisfies the lower bound constraint, it is considered valid, and we **output the solution**. If not, we proceed to Step 2.
> * Step 2: In the case where $k$ elements violate the lower bound constraint, based on Assumption 3.5, we set the final solution for these $k$ elements to the lower bound. Subsequently, we transform the remaining $N-k$ elements, creating **a new optimization problem with the same form in (5)**. We can then resolve this problem by repeating Step 1 and Step 2 iteratively.
>
> By employing Step 1, we attain the optimal solution for the unconstrained scenario. The definition of Assumption 3.5 enables us to deduce that the final solution will consistently be the optimal solution for equation (5) if the assumption holds true.
>
> **Lastly, we would like to express our gratitude for your patience in reviewing our response, and for your invaluable assistance in enhancing our paper thus far! Please let us know if you have any further questions. We are actively available!**

---

> > ### Comment · Reviewer_8NjX · 2023-08-17
> >
> > Thanks for the rebuttal! However, I am not sure whether it is reasonable to assume Assumption 3.5. In addition, I still feel the novelty of the algorithms is not very significant.  Therefore, I will retain my score.

---

> > > ### Author Response · Authors · 2023-08-17
> > >
> > > Thank you for the further comments! We will continue to revise and polish our work according to your advice. Your feedback is truly valuable to us!

---

### Official Review · Reviewer_n1BZ · 2023-07-28

**Soundness:** 3 good
**Presentation:** 3 good
**Contribution:** 2 fair
**Rating:** 4
**Confidence:** 3

**Summary:**

This paper focuses on constrained reinforcement learning and proposes a method called Constrained Proximal Policy Optimization (CPPO). Experimental results demonstrate the improved performance in terms of episodic return and episodic cost.


**Strengths:**

+ The proposed CPPO achieves improved balance between return and cost empirically.

**Weaknesses:**

- The novelty and contribution of this work is not clear. The difference between the proposed CPPO and CVPO does not seem to be very significant. Adopting the advantage value instead of Q-value is a natural extension. Besides, it is unclear how much sample complexity can be reduced by replacing $q$ by $v$. After all, $v$ should still satisfy the expectation constraint.
- The paper does not have any theoretical characterization of the proposed CPPO algorithm.


**Questions:**

- It is unclear why replacing Lagrangian based approach by the feasible region based approach can reduce the computational complexity. Solving a hard-constrained optimization problem seems to be more challenging. Please clarify.
- Can an off-policy version be developed for CPPO to improve its sampling efficiency?



**Limitations:**

 The paper mentions that CPPO method is an on-policy constrained RL, which suffers from lower sampling efficiency compared to other off-policy algorithm.

---

> ### Author Rebuttal · Authors · 2023-08-08
>
> **We would like to express our sincere gratitude to you for valuable and constructive comments, thanks for your support. Our responses are as follows.**
>
> **W1**: The novelty and contribution of this work is not clear. The difference between the proposed CPPO and CVPO does not seem to be very significant. Adopting the advantage value instead of Q-value is a natural extension. Besides, it is unclear how much sample complexity can be reduced by replacing $q$ by $v$. After all, $v$ should still satisfy the expectation constraint.
>
> **R1**: Thanks for your comment. We believe the proposed CPPO method is an extension of PPO in constrained RL field, rather than a straightforward on-policy CVPO approach. The only thing these two methods in common is that they share the same idea of finding an optimal policy within a trust region and progressively pushing the current policy towards the optimal one in an EM manner. Similar ideas have been successfully applied in CPO, MPO, and V-MPO as well.
>
> The most important contribution of our work is using the **probability ratio** instead of **probability density** to represent the optimal distribution. Previous MPO-based algorithms (MPO, CVPO, V-MPO) endeavour to directly derive the probability density $\psi$ according to $\int\psi(a|s)da=1$. However, note that this formulation does not inherently yield that $\sum\psi(a|s)=1$, **which is a error persists across all three aforementioned algorithms**, leading to the **incorrect normalization** of $\psi^*$. In contrast, our work addresses this issue by employing the probability ratio $v$, which allows for a more straightforward calculation of the distribution with ensuring $E(v)=1$.
>
> Furthermore, in contrast to the KL divergence, the utilization of the $l_2$-norm ($\chi^2$ divergence) during the E-step offers a distinct **geometric interpretation** of the feasible region. This perspective facilitates the formulation of the **recovery update method**, which effectively minimizes costs while maintaining rewards.  In M-step, the proposed CPPO algorithm conduct a policy update process akin to that of PPO, thereby obviating the necessity for additional hyperparameters as seen in other MPO-based approaches.
>
> For the sample complexity, CPPO efficiently estimates the cost return by combining the surrogate cost objective (calculated from the **cost advantage value**) with the **cost return of the current policy**. Unlike CVPO, which necessitates sampling several actions under the same state, CPPO avoids this requirement, making it with less sample complexity.
>
> **W2**: The paper does not have any theoretical characterization of the proposed CPPO algorithm.
>
> **R2**: Thanks for your comment. Conducting the convergence analysis of CPPO method is a challenging work and it is hard to address this issue in the current paper. We will look into adding convergence analysis for the CPPO algorithm in our future work, if possible.
>
>
> **Q1**: It is unclear why replacing Lagrangian based approach by the feasible region based approach can reduce the computational complexity. Solving a hard-constrained optimization problem seems to be more challenging. Please clarify.
>
> **A1**: Thanks for your comment. As mentioned in the Introduction, the advantage of the feasible region-based approach over the primal-dual method is its superior **convergence speed**. The feasible region method implicitly determines the Lagrange multiplier, eliminating the need for updating it through the gradient. The experimental results in the Circle environment and the comparison[1] between CPO and PDO[2] also further validate this advantage.
>
> **Q2**:Can an off-policy version be developed for CPPO to improve its sampling efficiency?
>
> **A2**: Thanks for your comment. We are in the process of developing the off-policy CPPO algorithm, wherein we leverage sampled trajectories from the trajectories buffer. This method bears similarities to ACER[3]. We believe that this approach holds promise and can potentially address certain challenges in our research domain
>
>
>
> **Lastly, we would like to express our gratitude for your patience in reviewing our response, and for your invaluable assistance in enhancing our paper thus far! Please let us know if you have any further questions. We are actively available!**
>
> \[1\] Achiam, J., Held, D., Tamar, A., & Abbeel, P. (2017, July). \"Constrained policy optimization\". In International conference on machine learning (pp. 22-31). PMLR.
>
> \[2\] Chow, Yinlam, et al. \"Risk-constrained reinforcement learning with percentile risk criteria.\" The Journal of Machine Learning Research 18.1 (2017): 6070-6120.
>
> \[3\] Wang, Ziyu, et al. "Sample efficient actor-critic with experience replay." arXiv preprint arXiv:1611.01224 (2016).

---

> > ### Comment · Reviewer_n1BZ · 2023-08-18
> >
> > Thanks for the rebuttal. After going through the reviews and rebuttals, I feel the novelty and contribution of this work need to be strengthened in order to get it accepted. I have decided to keep my original rating.

---

> > > ### Author Response · Authors · 2023-08-18
> > >
> > > Thank you for the further comments! We will continue to revise and polish our work according to your advice. Your feedback is truly valuable to us!

---

### Author Rebuttal · Authors · 2023-08-09

**We would like to express our sincere gratitude to reviewers for valuable and constructive comments.**

Many reviewers have raised questions regarding the novelty of our work. It is important to emphasize that the proposed CPPO method is an extension of PPO in constrained RL field, rather than a straightforward on-policy CVPO approach, despite they shares the same  expectation-maximization principle. The most important difference and contribution lies on the utilization of the **probability ratio** instead of **probability density** to represent the optimal distribution. This modification not only simplifies the computation of the optimal distribution but also lays the foundation for subsequent geometric interpretations and the development of the recovery update strategy.

**Lastly, we would like to express our gratitude for reviewers' patience in reviewing our response, and for reviewers' invaluable assistance in enhancing our paper thus far! Please let us know if you have any further questions. We are actively available!**

---

### Decision · Program_Chairs · 2023-09-21

**Decision:**

Reject

**Comment:**

This work presents a new approach to constrained reinforcement learning by introducing Constrained Proximal Policy Optimization (CPPO), a specialized version of PPO designed for solving Markov Decision Process (MDP) problems in a discounted reward setting with discounted cost constraints. While the experimental results show improvements in terms of episodic return and episodic cost, several critical issues remain to be addressed.
1. The paper lacks a comprehensive literature review, which is essential for situating this work within the broader context of existing research in constrained reinforcement learning and MDPs. A more thorough survey would validate the novelty of the work and help identify gaps that this research fills.
2. The absence of a convergence analysis is a significant drawback. Convergence analysis is crucial for understanding the stability and reliability of the proposed CPPO technique, especially given its application in constrained settings. This is also because the other existing works in the literature have provided in-depth analysis of the proposed techniques for CMPD as highlighted by the reviewers.
3. The authors' response to the concerns raised by the reviewer was not satisfactory. Specifically, questions about the lack of convergence analysis and the use of Assumption 3.5 in the paper were not adequately addressed.
4. After reviewing the rebuttal, it appears that the contributions of this work are limited, requiring the paper to undergo careful revision, particularly around the theoretical analysis of the proposed CPPO technique.

In summary, while the paper makes a valuable initial contribution by introducing the CPPO method for constrained MDPs, it falls short in critical areas that are essential for a well-rounded scientific paper. These include needing a more comprehensive literature review a detailed convergence analysis. Addressing these issues would significantly enhance the quality of the paper.